# Sulfide-Responsive Transcription Control in *Escherichia coli*

**DOI:** 10.3390/microorganisms13020344

**Published:** 2025-02-05

**Authors:** Koichi Hori, Rajalakshmi Balasubramanian, Shinji Masuda

**Affiliations:** 1School of Life Science & Technology, Tokyo Institute of Technology, Yokohama 226-8501, Japan; 2School of Life Science & Technology, Institute of Science Tokyo, Yokohama 226-8501, Japan

**Keywords:** antibiotic tolerance, *Escherichia coli*, hydrogen sulfide, persulfide, reactive sulfur species, supersulfide, YgaV

## Abstract

To elucidate the mechanism of large-scale transcriptional changes dependent on sulfide in *Escherichia coli*, a large-scale RNA-sequencing analysis was performed on wild-type and sulfide-responsive transcription factor YgaV deletion mutants grown under three conditions: aerobic, semi-aerobic, and semi-aerobic with sulfide. The resulting dataset from these six conditions was subjected to principal component analysis, which categorized the data into five principal components. Estimation of the typical gene expression regulatory mechanisms in each category suggested the presence of mechanisms that are dependent on sulfide but independent of YgaV, as well as those that depend on YgaV but not on sulfide. In YgaV-dependent transcriptional regulation, YgaV was found to function as both a repressor and an activator. These results support the previous hypothesis that YgaV acts as a global regulator responsible for redox homeostasis.

## 1. Introduction

In the gut microbiota, over 100 bacterial species metabolize sulfur components within host cells [1,2]. As a result, the extracellular sulfide concentration in the intestine fluctuates between approximately 0.2 to 2.4 mM [1,3]. In such an environment, oxygen consistently acts as a limiting factor for ATP production through respiration [4]. Under these oxygen-limited conditions, microorganisms perform anaerobic respiration using alternative electron acceptors such as fumarate, nitrate, and nitrite. Intestinal bacteria frequently switch between aerobic and anaerobic growth modes, as well as conditions of high and low sulfide concentrations, to facilitate their survival in the gastrointestinal environment of the host [4]. In *Escherichia coli*, intracellular sulfide levels are also regulated through various metabolic pathways. For instance, 3-mercaptopyruvate sulfurtransferase and cysteine desulfurase are responsible for the majority of endogenous H_2_S production in *E. coli*, contributing to the suppression of reactive oxygen species formation and protection from antibiotic stress through the regulation of iron homeostasis [5,6]. Intracellular sulfide in bacteria is further converted into more oxidized sulfane sulfur species, known as supersulfides, through metabolic processes [3,7,8,9,10,11,12,13,14].

Identified transcriptional regulators that specifically respond to sulfide/supersulfide are categorized into three groups based on structural differences: the arsenic repressor (ArsR), the copper-sensitive operon repressor (CsoR), and the Fis superfamily [15,16,17,18,19]. Previously, we identified the ArsR-type sulfide-responsive transcriptional regulator SqrR in the purple photosynthetic bacterium *Rhodobacter capsulatus* and characterized its properties [17]. SqrR controls gene expression in response to the availability of H_2_S, which is essential for photoautotrophic growth, regulation of biofilm formation and gene transfer, as well as modulating c-di-GMP signaling [17,20]. Specifically, when extracellular H_2_S becomes available, two conserved cysteine residues in SqrR react with supersulfide to form an intramolecular tetrasulfide bond, leading to the de-repression of photosynthesis gene expression [17,21,22], which could be reduced by thioredoxin [23]. Interestingly, homologs of SqrR are widely conserved in non-sulfur bacteria, including *E. coli* [19]. Characterization of YgaV, the SqrR homolog in *E. coli*, revealed that YgaV suppresses the transcription of genes involved in anaerobic respiration in the absence of sulfide and activates the expression of genes related to iron uptake in the presence of sulfide [9,24]. Both SqrR and YgaV can bind to heme, which may influence their DNA-binding affinity [9,25]. YgaV-dependent transcriptional regulation prevents reactive oxygen species production, contributing to oxidative stress response and antibiotic resistance [9,24]. However, many aspects of the YgaV-dependent transcriptional regulation mechanism and the transcriptional regulatory network involving YgaV remain poorly understood.

In this study, we performed RNA sequencing (RNA-seq) analysis on wild-type and *ygaV* mutant *E. coli* cultured under various conditions to identify the large-scale transcriptional regulation based on sulfide concentration and the role of the sulfide-responsive transcription factor YgaV in this regulation. This comprehensive transcriptomics approach should provide an overview not only of the features of YgaV-dependent transcriptional activation and repression but also of YgaV-independent and sulfide-dependent transcriptional regulation. The results revealed that YgaV positively or negatively regulates large-scale gene expression in response to sulfide.

## 2. Materials and Methods

### 2.1. Bacterial Strains and Growth Conditions

*Escherichia coli* BW25113 [*rrnB*, DE*lacZ*4787, *HsdR*514, DE (*araBAD*)567, DE (*rhaBAD*)568, *rph−1*] was used as the WT, which was the parent strain of the KEIO mutant collection [26,27]. The *ygaV* mutant (Δ*ygaV*) was previously constructed from the BW25113 strain [9]. *E. coli* strains were cultured in LB medium at 37 °C. For aerobic growth, 3 mL LB medium was shacked vigorously in a test tube (φ19 mm). For semi-aerobic growth, 7 mL of LB medium was cultured in a screw-capped glass tube (φ10 mm × 100 mm) with gentle shacking. 0.2 mM Na_2_S (final concentration) was added when necessary.

### 2.2. Sample Preparation for RNA-Seq and Data Analysis

*E. coli* WT and Δ*ygaV* mutants were grown overnight in LB medium at 37 °C. Next day, each culture was diluted 100 times in test tubes for aerobic growth and 20 times in glass tubes for semi-aerobic growth, respectively, using fresh LB medium. For aerobic growth cultures, we harvested cells during the early to mid-log phase (OD600 values between 0.25 and 0.30) to prevent oxygen limitation in the medium (Appendix A) [9]. The doubling time of the strains under aerobic conditions was approximately 33 min (Appendix A). For semi-aerobic growth cultures, cells were harvested during the mid to late-log phase (OD600 values between 0.5 and 0.6). The doubling time of all strains under semi-aerobic conditions was approximately 119 min (Appendix A). One milliliter of each culture was then harvested by centrifugation at 5000× *g* and temperature of 4 °C for 1 min. Total RNAs were subsequently extracted using the RNeasy Mini kit (Qiagen, Hilden, Germany), followed by TURBO DNase (Ambino, Waltham, MA, USA) treatment to eliminate genomic DNA according to manufacturer’s instructions, after which an equal quantity of total RNAs from three biologically independent samples of each strain was mixed for use in RNA sequencing. RIKEN Genesis, Japan conducted the subsequent RNA-sequencing procedures. The libraries were constructed by the TruSeq Stranded total RNA kit (Illumina, San Diego, CA, USA) with NEBNext Bacterial rRNA depletion Kit (NEB, Ipswich, MA, USA) and 150 bp paired-end sequencing was performed on a NovaSeq 6000 (Illumina). The sequencing data were deposited in DDBJ Sequence Read Archive under the accession numbers DRR609299-DRR609302 (BioProject: PRJDB12324).

RNA-seq reads were mapped and counted for the BW25113 (GCA_000750555.1) chromosome using HISAT2 [28]. Note that data for WT and Δ*ygaV* grown under AC conditions were reproduced from our previous study (DRR318756, DRR318757). Normalization of read counts and detection of DEGs (*p*-value < 0.05) were calculated with iDEGES/deseq pipeline using the TCC package (The R version 4.1.2, 1.34.0) [29]. Normalized read counts across all samples were shown in Appendix A and used to draw heat maps. Normalized read counts between each sample were used to detect DEGs and to draw volcano plot. PCA of normalized counts (Log2) was performed using the prcomp R package (The R version 4.1.2, The R Stats Package version 4.1.2). KEGG enrichment analysis was implemented in KAAS [https://www.genome.jp/kegg/kaas/ (accessed on 20 June 2020)] [30] and KEGG Mapper reconstruct tool [https://www.genome.jp/kegg/mapper/reconstruct.html (accessed on 20 June 2020)] [31].

## 3. Results and Discussion

We previously performed RNA-seq analysis of *E. coli* WT and Δ*ygaV* mutant grown under aerobic (AC) conditions [9]. The results indicated that anaerobic respiratory genes, such as nitrate and nitrite reductase genes, are upregulated in Δ*ygaV* mutant, suggesting that YgaV-dependent transcriptional regulation is related to growth under oxygen-limiting conditions. In the gut, oxygen levels are consistently low, and sulfide levels vary under such conditions. To evaluate the importance of YgaV function in *E. coli* growth under these gut-like conditions, we conducted RNA-seq of *E. coli* WT and Δ*ygaV* grown under semi-aerobic (SA) conditions with and without sulfide (0.2 mM Na_2_S). Notably, treatment with 0.2 mM Na_2_S can induce the depression of the YgaV-target gene expression in vivo [9].

We found that 74 genes were significantly upregulated, and 22 genes were significantly downregulated (*p*-value < 0.05) in WT cells grown under SA conditions compared to those grown under AC conditions (Appendix A). The upregulated genes include those involved in primary and secondary metabolism, carbohydrate metabolism, energy metabolism, and amino acid metabolism, as estimated by the Kyoto Encyclopedia of Genes and Genomes (KEGG) database (Appendix A), suggesting that oxygen limitation significantly affects gene expression profiles globally.

Under SA conditions, 132 genes were significantly downregulated (*p*-value < 0.05) in WT cells grown with sulfide compared to those grown without sulfide (Appendix A), indicating that sulfide suppresses gene expression related to energy and primary metabolisms. However, the overall changes in gene expression between sulfide-treated and sulfide untreated conditions in the Δ*ygaV* mutant were less pronounced than in WT cells, and there was no correlation (Appendix A), suggesting that YgaV serves as a key regulator in the sulfide response. In the Δ*ygaV* mutant, compared to the WT, 224, 223, and 172 genes were identified as significantly differentially expressed (*p*-value < 0.05) under AC, SA without sulfide, and SA with sulfide conditions, respectively (Appendix A). Notably, many of these differentially expressed genes (DEGs) were unique to each condition. These results indicate that YgaV regulates distinct targets under AC and SA conditions without sulfide conditions, as well as in response to sulfide.

The obtained RNA-seq data were further analyzed using principal component (PC) analysis to gain deeper insights into the YgaV function. The results indicated that nearly all the variance in the data could be explained by five components: PC1 (72.1%), PC2 (17.7%), PC3 (5.3%), PC4 (3.1%), and PC5 (1.8%) (Figure 1). The PC scores for each sample were calculated by summing the values obtained from multiplying each expression level by its corresponding eigenvector (Appendix A). PC1 separated the oxygen levels in the growth conditions (AC or SA), independent of genotype (WT or Δ*ygaV*) (Figure 1). In PC2, WT grown under SA conditions with sulfide was distinct from other conditions. The differences in PC3 scores for the Δ*ygaV* mutant under each condition were greater than those of WT in the corresponding environmental conditions. Under SA conditions, there were also remarkable differences in PC3 scores between WT and the Δ*ygaV* mutant. In PC4, WT grown under SA conditions and the Δ*ygaV* mutant grown under SA conditions with sulfide were separated in opposite directions from the other samples. In PC5, WT and Δ*ygaV* mutants grown under AC conditions were separated in different directions from the rest (Figure 1). Expression changes in genes with high or low eigenvector contribute to the spread of PC scores. Figure 2 shows heat maps for the expression levels of the top 100 genes with the highest and lowest eigenvectors in each PC (High_100 and Low_100, respectively).

Under AC conditions, genes with the highest PC1 eigenvector values were downregulated, while those with the lowest PC1 eigenvector values were upregulated, regardless of loss of YgaV (Figure 2). Figure 3 shows the expression profiles (read counts from RNA-seq) and eigenvector values of *bssR* and *fecB*, each representing typical expression profiles of the top 100 highest and lowest eigenvector genes in PC1, respectively. Transcript levels of *bssR* and *fecB* in WT and the Δ*ygaV* mutant grown under AC conditions were lower and higher, respectively, compared to other conditions. This suggests that YgaV does not influence the expression of these genes, indicating the involvement of other regulatory mechanisms in their transcriptional control. We proposed models to explain the expression profiles of these groups. Under AC conditions, RNA polymerase (RNAP), with or without transcription factors, is inactive for genes in the highest eigenvector group (Figure 3, top right) and active for genes in the lowest eigenvector group (Figure 3, bottom right). Conversely, under SA conditions, RNAP is active for genes in the highest PC1 group (Figure 3, top right) and inactive for those in the lowest PC1 group (Figure 3, bottom right). We hypothesize that RNAP activity is inhibited by sulfide for transcription of genes in the lowest PC1 group through an unknown mechanism.

Figure 4 shows the expression profiles and eigenvector values of *ebgR* and *dppD*, each representing typical expression profiles of the top 100 highest and lowest eigenvector genes in PC2, respectively. Transcript levels of *ebgR* and *dppD* in WT grown under SA conditions with sulfide were lower and higher, respectively, compared to other conditions. This pattern is also observed in genes with the highest or lowest PC2 eigenvector values, respectively. The sulfide-dependent expression changes are abolished in the Δ*ygaV* mutant (Figure 2), suggesting that these changes are regulated by YgaV. We propose models to explain these expression profiles. Under SA conditions, sulfide catalyzes the formation of tetra-sulfide bonds in YgaV, enhancing its DNA-binding affinity to promoters of genes with the highest (Figure 4, top right) and lowest PC2 eigenvectors (Figure 4, bottom right). The modified YgaV appears to activate the transcription of genes in the highest eigenvector group (Figure 4, top right) and inhibit the transcription of genes in the lowest eigenvector group (Figure 4, bottom right).

Although genes with the highest PC3 eigenvector values were upregulated by sulfide in the Δ*ygaV* mutant, these genes originally showed higher expression levels in the WT than in the Δ*ygaV* mutant under SA without sulfide (Figure 2). Conversely, genes with the lowest PC3 eigenvector, which were highly expressed in the Δ*ygaV* mutant, were suppressed by sulfide to match the expression levels seen in the WT (Figure 2). Figure 5 shows the expression profiles and eigenvector values of *icdC* and *yggX*, representing typical expression profiles of the top 100 highest and lowest eigenvector genes in PC3, respectively. Transcript levels of *icdC* in WT grown under SA conditions without sulfide and in the Δ*ygaV* mutant grown under SA conditions with sulfide were higher than those in other conditions. Conversely, transcript levels of *yggX* in the Δ*ygaV* mutant grown under SA conditions without sulfide were higher than those in other conditions. We propose models to explain these expression profiles. Under SA sulfide-free conditions, basal activity of RNAP is inactive for genes with the highest eigenvector (Figure 5, top right) and active for those with the lowest eigenvector (Figure 5, bottom right), with YgaV functioning as an activator for the former transcription (Figure 5, top right) and as a repressor for the latter transcription (Figure 5, bottom right). In the presence of sulfide, RNAP is active for genes in the highest eigenvector group (Figure 5, top right) and inactive for those in the lowest eigenvector group (Figure 5, bottom right), with sulfide-modified YgaV acting as a repressor for the transcription of the highest eigenvector genes (Figure 5, top right).

The expression levels of genes with the highest and lowest PC4 eigenvector values were distinct in the WT grown under SA conditions without sulfide. Compared to other SA conditions, genes with the highest PC4 eigenvector were generally expressed at lower levels, while genes with lowest PC4 eigenvector were generally expressed at higher levels in the WT grown under SA conditions without sulfide (Figure 2). Figure 6 shows the expression profiles and eigenvector values of *modA* and *intZ*, each representing typical expression profiles of the top 100 highest and lowest eigenvector genes in PC4, respectively. Transcript levels of *modA* in the Δ*ygaV* mutant grown under SA conditions with and without sulfide were higher than those in other conditions. While transcript levels of *intZ* in WT grown under SA conditions without sulfide were higher than in other conditions. We propose models to explain these expression profiles. Under SA conditions with or without sulfide, basal activity of RNAP is active for genes in the highest eigenvector group (Figure 6, top right) and inactive for the lowest eigenvector group (Figure 6, bottom right). Although YgaV acts as a repressor for the transcription of the highest eigenvector genes (Figure 6, top right) and as an activator for the transcription of the lowest eigenvector genes (Figure 6, bottom right), sulfide-modified YgaV loses these transcriptional regulatory functions (Figure 6, right).

Distinct expression patterns in PC5, which do not overlap with those of PC1-4, were observed (Figure 2). Under AC conditions, the deletion of YgaV resulted in decreased expression levels of genes with the highest PC5 eigenvector and increased expression levels of genes with the lowest PC5 eigenvectors. However, these expression changes were not observed under SA conditions (Figure 2). Figure 7 shows the expression profiles and eigenvector values of *mglB* and *tdcB*, each representing typical expression profiles of the top 100 highest and lowest eigenvector genes in PC5, respectively. Transcript levels of *mglB* in the Δ*ygaV* mutant were lower than those in WT grown under AC conditions. Conversely, transcript levels of *tdcB* in the Δ*ygaV* mutant were higher than those in WT grown under AC conditions. We propose models to explain these expression profiles. Under AC conditions, RNAP, with or without transcription factors other than YgaV, is inactive for genes in the highest eigenvector groups (Figure 7, top right) and active for genes in the lowest eigenvector groups (Figure 7, bottom right). However, unmodified YgaV which binds heme under AC conditions functions as an activator for the former group (Figure 7, top right) and repressor for the latter group (Figure 7, bottom right).

## 4. Conclusions

Our RNA-seq analysis reveals that the mutational loss of *ygaV* induces drastic changes in gene expression profiles under oxygen-limiting and/or sulfide-enriched conditions (Figure 1 and Figure 2), confirming the crucial role of YgaV as a global regulator of gene expression required for redox homeostasis in *E. coli*. The complex regulatory mechanisms of YgaV-dependent transcriptional regulation suggest that both unmodified YgaV and sulfide-modified YgaV can function as either a repressor or an activator, which is consistent with our previous studies. Specifically, under sulfide free AC conditions, YgaV represses transcription of anaerobic respiratory genes, such as those encoding dimethyl sulfoxide reductase, nitrite reductase, and cytochrome *bd* oxidase. Given that sulfide inactivates the activity of cytochrome *bo*-type terminal oxidase [32], this repression appears to be critical for preventing the generation of reactive-oxygen-species, which are important for antibiotic tolerance [9]. Recently, we demonstrated that YgaV upregulates (rather than derepresses) genes involved in iron uptake under sulfide-stress conditions [24]. Since adequate levels of iron are required to scavenge reactive oxygen species, while excess iron catalyzes generation of reactive oxygen species, this highlights the importance of YgaV’s dual function in maintaining redox homeostasis.

Furthermore, OxyR, previously recognized as a reactive oxygen species-specific transcription factor, was recently shown to sense supersulfide [33]. This suggests that YgaV and OxyR work together to enhance dynamics of transcriptional regulation of target genes, thereby helping to prevent reactive-oxygen-species generation. The coordinated transcriptional regulation by YgaV and OxyR in response to sulfide and reactive oxygen species may play a vital role for bacteria inhabiting the gut, where sulfide, iron, and oxygen levels vary significantly.

## Figures and Tables

**Figure 1 microorganisms-13-00344-f001:**
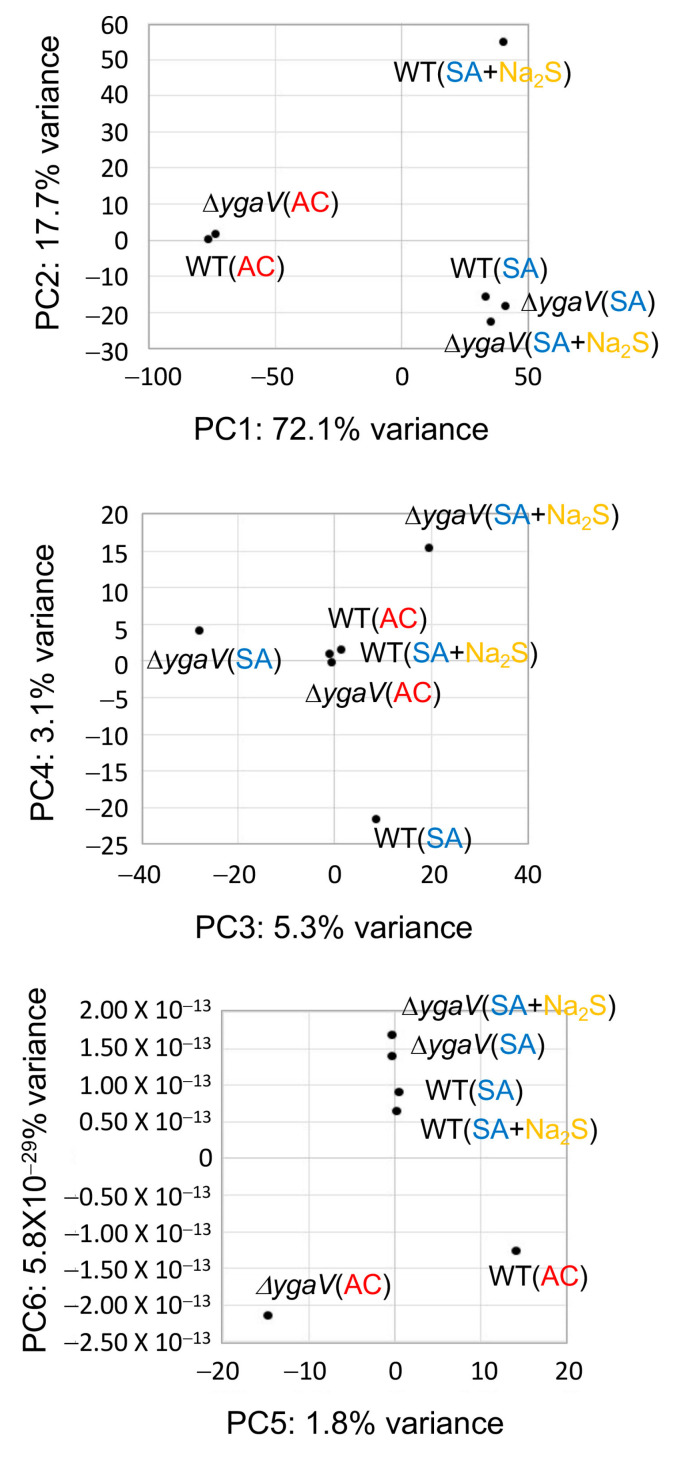
Principal component analysis score plots of RNA seq data from WT and Δ*ygaV* mutant grown under aerobic (AC) and semi-aerobic (SA) conditions with (+Na_2_S) or without 0.2 mM Na_2_S. The percentage of variation explained by each PC is given within axis labels.

**Figure 2 microorganisms-13-00344-f002:**
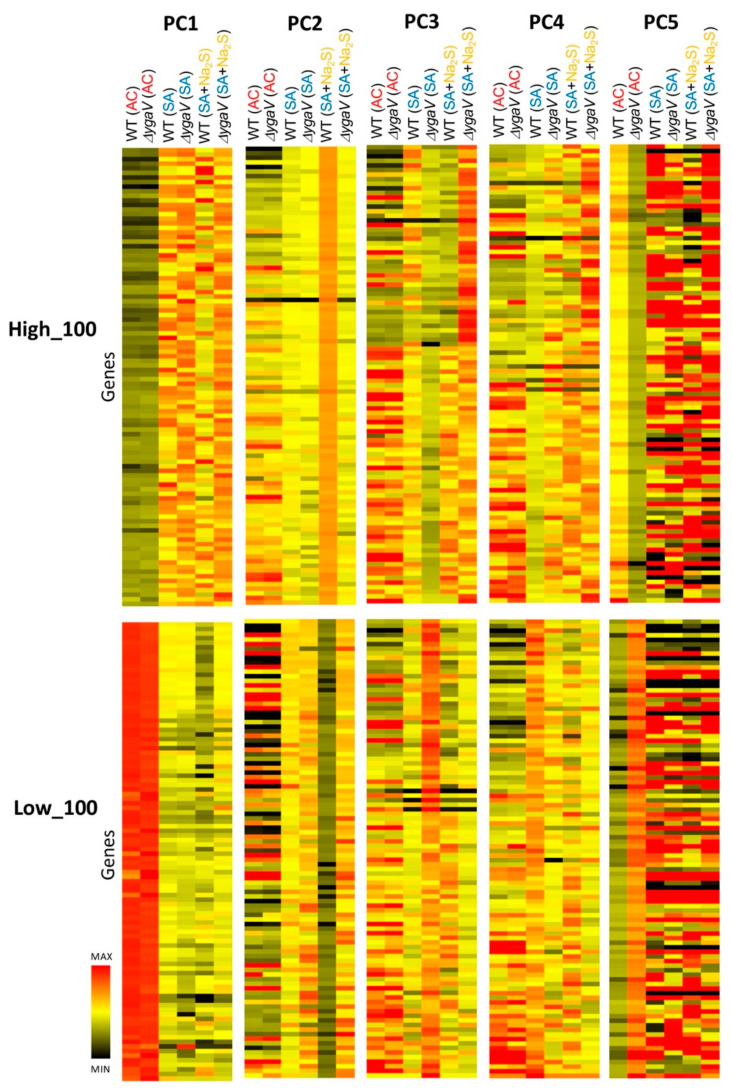
Heatmaps depicting the expression levels of the top 100 genes in each component (PC1 to PC5). The relative expression level (log2) of each gene is represented by a color gradient, with lighter colors indicating higher expression and darker colors indicating lower expression.

**Figure 3 microorganisms-13-00344-f003:**
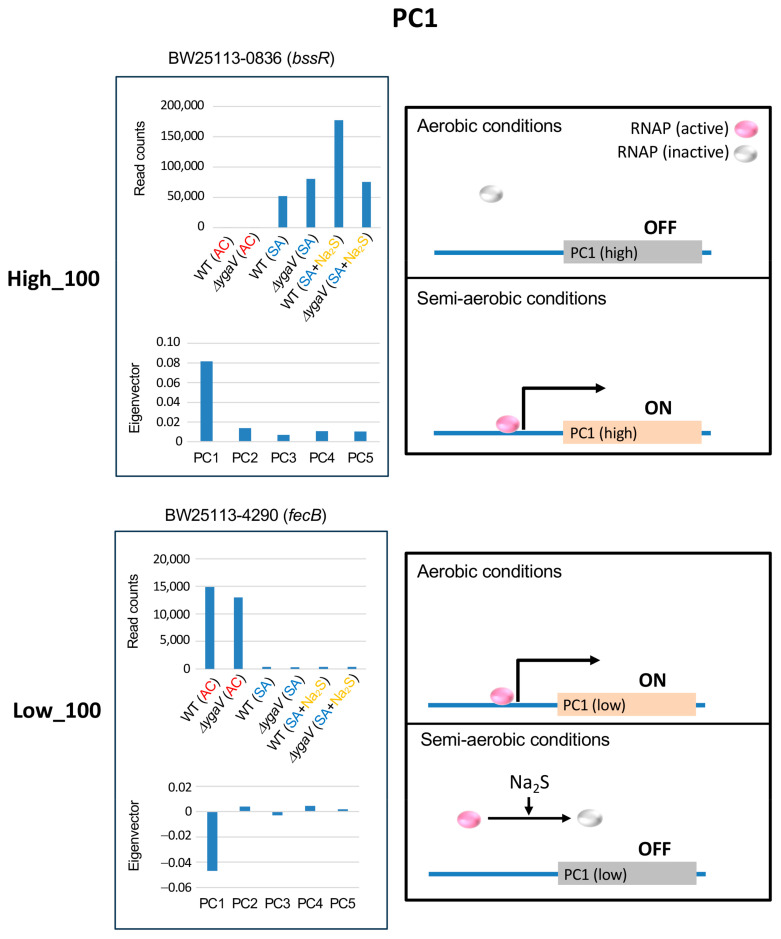
Model for YgaV-dependent regulation of the top 100 genes with the highest (High_100) and lowest (Low_100) eigenvector values in PC1. Expression profiles of *bssR* and *fecB* were shown as examples (**left**). The action of RNA polymerase (RNAP) in cells under aerobic and semi-aerobic conditions is illustrated (**right**).

**Figure 4 microorganisms-13-00344-f004:**
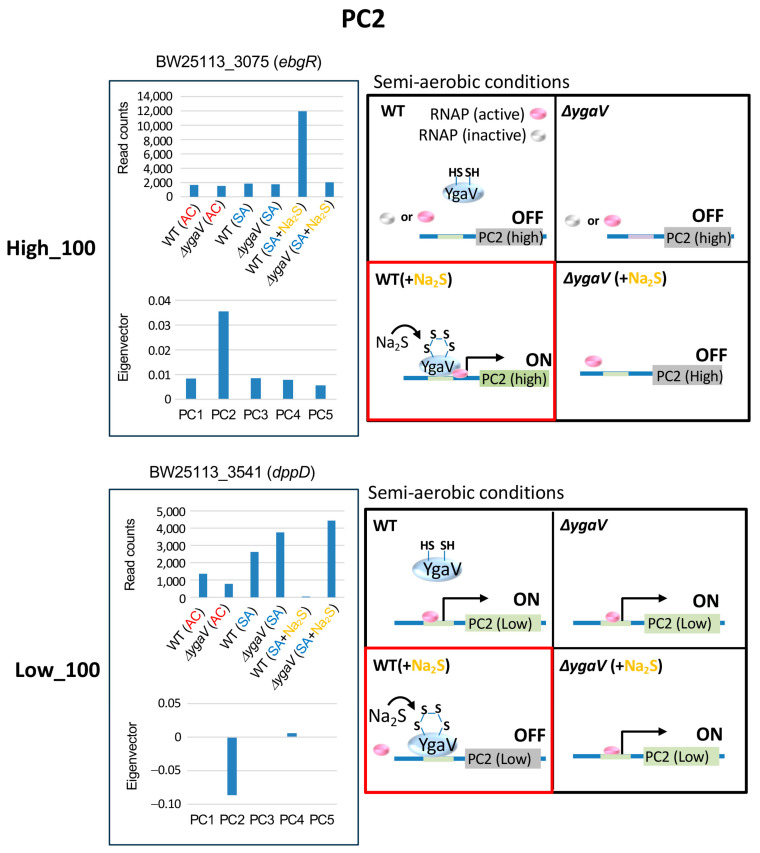
Model for YgaV-dependent regulation of the top 100 genes with the highest (High_100) and lowest (Low_100) eigenvector values in PC2. Expression profiles of *ebgR* and *dppD* were shown as examples (**left**). The actions of RNA polymerase (RNAP) and YgaV in cells under semi-aerobic conditions with and without sulfide (Na_2_S) are illustrated (**right**). Red squares represent the modes of action of YgaV as discussed in the text.

**Figure 5 microorganisms-13-00344-f005:**
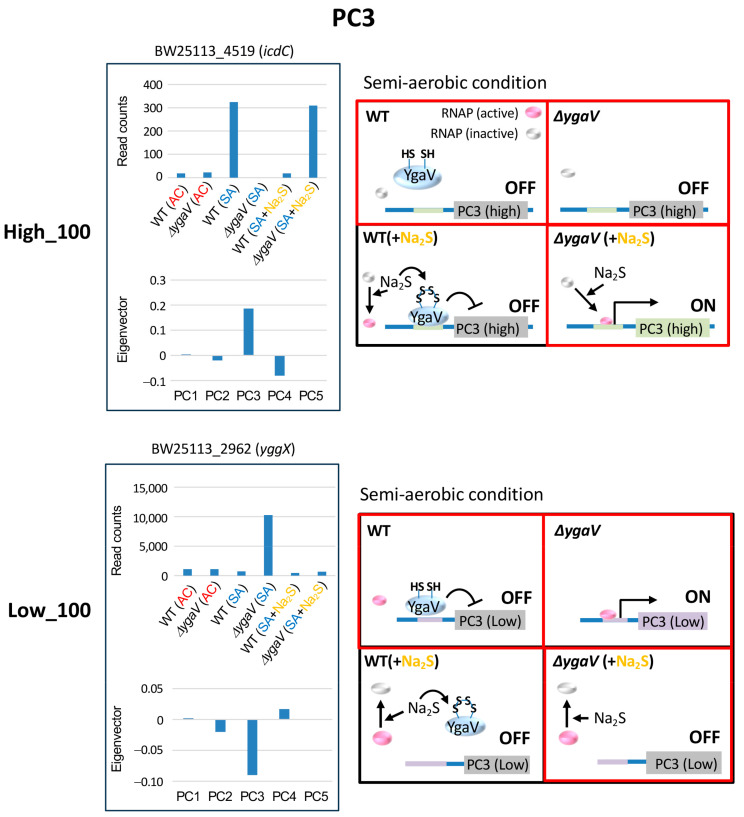
Model for YgaV-dependent regulation of the top 100 genes with the highest (High_100) and lowest (Low_100) eigenvector values in PC3. Expression profiles of *icdC* and *yggX* were shown as examples (**left**). The actions of RNA polymerase (RNAP) and YgaV in cells under semi-aerobic conditions with and without sulfide (Na_2_S) are illustrated (**right**). Red squares represent the modes of action of YgaV as discussed in the text.

**Figure 6 microorganisms-13-00344-f006:**
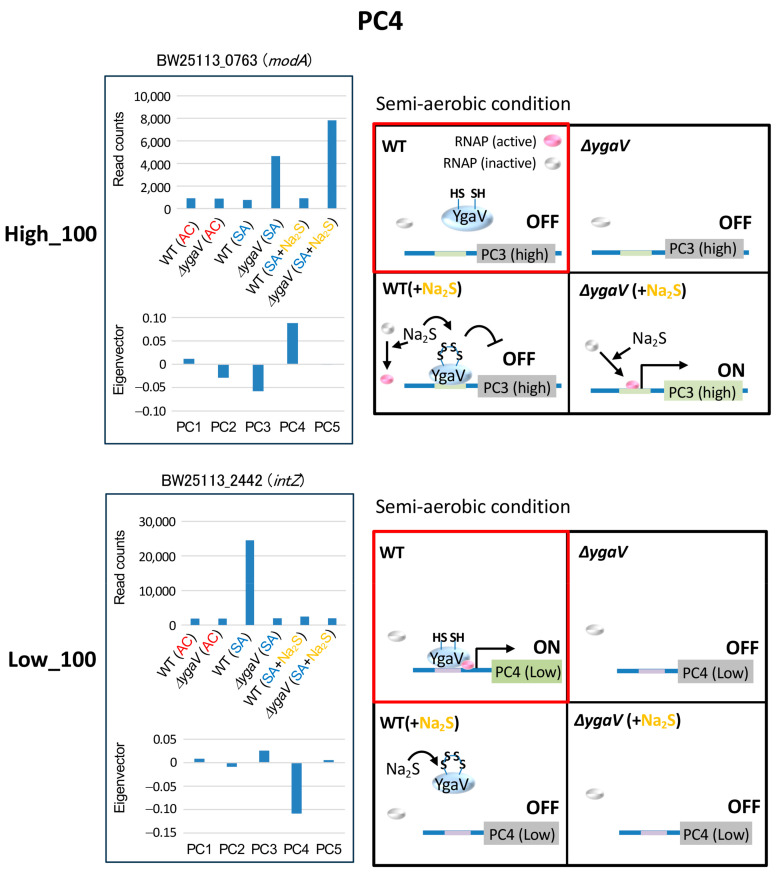
Model for YgaV-dependent regulation of the top 100 genes with the highest (High_100) and lowest (Low_100) eigenvector values in PC4. Expression profiles of *modA* and *intZ* were shown as examples (**left**). The actions of RNA polymerase (RNAP) and YgaV in cells under semi-aerobic conditions with and without sulfide (Na_2_S) are illustrated (**right**). Red squares represent the modes of action of YgaV as discussed in the text.

**Figure 7 microorganisms-13-00344-f007:**
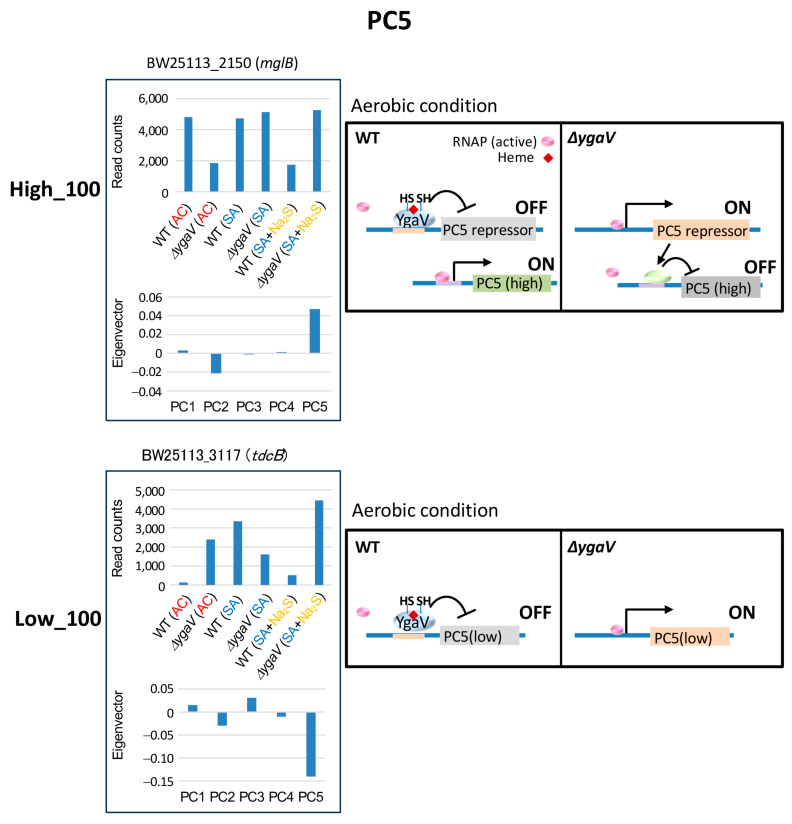
Model for YgaV-dependent regulation of the top 100 genes with the highest (High_100) and lowest (Low_100) eigenvector values in PC5. Expression profiles of *mglB* and *tdcB* were shown as examples (**left**). The actions of RNA polymerase (RNAP) and YgaV in cells under aerobic conditions are illustrated (**right**).

## Data Availability

The original contributions presented in this study are included in the article/Appendix A. Further inquiries can be directed to the corresponding author.

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
