# Peer review of "Sulfide-Responsive Transcription Control in Escherichia coli"

_microorganisms, 2025, doi:10.3390/microorganisms13020344_

Round 1

Reviewer 1 Report

Comments and Suggestions for Authors

The authors attempted to identify the mechanism of sulfide-dependent large-scale transcriptional changes in E. coli by performing large-scale RNA sequencing analysis of wild-type and sulfide-responsive transcription factor YgaV deletion mutants under aerobic, semiaerobic, and semiaerobic plus sulfide conditions. There are the following suggestions.

“Introduction” section

1.       Does YgaV affect antibiotic resistance in bacteria?

“Materials and methods” section

2.       Line 75, For semi-aerobic growth, how to make semi-aerobic condition? Full the medium? More details should be explained.

3.       Lines 79-80, “Next, day,” should be “Next day,” .

4.       Line 82, the time of centrifugation should mentioned.

5.       Lines 85-86, “total RNAs from three biologically independent samples of each strain was mixed for use in RNA sequencing.” Does it mean that all samples are mixed for one time RNA sequencing? Not repeated three times RNA sequencing?

“Results and discussion”

6.       Does YgaV affect the expression of iron-sulfur proteins? E.g. An important protein LipA for glycolysis, which containing a species consistent with an S = 0 [4Fe-4S](2+) cluster(ref: PMID: 28068366PMID: 11106496)? try to show some data in results or discussion.

“Reference”

7.       Consistent all references. E.g. References 1 and 2 are different with others.

Author Response

Thank you very much for your thoughtful comments and recommendations. We revised the manuscript based on your comments and recommendations. Specific points are below. Please note that changes are highlighted in red in the revised manuscript.

Reviewer 1

The authors attempted to identify the mechanism of sulfide-dependent large-scale transcriptional changes in E. coli by performing large-scale RNA sequencing analysis of wild-type and sulfide-responsive transcription factor YgaV deletion mutants under aerobic, semiaerobic, and semiaerobic plus sulfide conditions. There are the following suggestions.

“Introduction” section

Comment 1: Does YgaV affect antibiotic resistance in bacteria?

Response 1: Yes. In the revised manuscript, we carefully discuss the possible mechanisms of YgaV for antibiotic tolerance (Page 286~303).

“Materials and methods” section

Comment 2: Line 75, For semi-aerobic growth, how to make semi-aerobic condition? Full the medium? More details should be explained.

Response 2: We monitored the growth of the strains by measuring their growth curves under aerobic and semi-aerobic conditions (with and without sulfide) (Pages 86–93). The growth curves have been newly included in the supplemental figure (Supplemental Fig. S1), which shows that the doubling time of the strains under aerobic conditions was approximately 33 minutes. In contrast, the doubling time under semi-aerobic conditions (both with and without sulfide) was approximately 119 minutes. These results indicate that the growth of the strains was slowed by oxygen limitation under semi-aerobic conditions.

Comment 3:  Lines 79-80, “Next, day,” should be “Next day,”

Response 3: Thank you for pointing out this. We corrected in the revised manuscript (Page 84~85).

Comment 4: Line 82, the time of centrifugation should mentioned.

Response 4: Thank you for pointing out this. We included time of centrifugation in the revised manuscript (Page 94).

Comment 5: Lines 85-86, “total RNAs from three biologically independent samples of each strain was mixed for use in RNA sequencing.” Does it mean that all samples are mixed for one time RNA sequencing? Not repeated three times RNA sequencing?

Response 5: Yes. We did such a way. This is because we performed the same RNA seq analysis previously, so that we did the experiment in the same way to comparison analysis.

“Results and discussion”

Comment 6: Does YgaV affect the expression of iron-sulfur proteins? E.g. An important protein LipA for glycolysis, which containing a species consistent with an S = 0 [4Fe-4S](2+) cluster(ref: PMID: 28068366,PMID: 11106496)? try to show some data in results or discussion.

Response 6: Thank you for the information. YgaV controls iron uptake gene expression perhaps independent to Fur. This suggest that YgaV controls directly and/or indirectly controls iron-sulfur proteins at least assembly of the clusters. However, because we hope that this study forcus on more conprehesive overview of transcriptional changes in response to sulfide, specific gene expression analysis in specific function would remain for future studies. We would like to perform further experiments in future from the view point of iron-sulfur cluster-containing proteins and kindly ask the reviewer to understand our reasoning. Thank you for the suggestion

“Reference”

Comment 7:  Consistent all references. E.g. References 1 and 2 are different with others.

Response 7: Thank you for pointing out this. We correct all in the revised manuscript.

Reviewer 2 Report

Comments and Suggestions for Authors

The results of the study may be used for understanding global expression regulation mechanisms in bacteria.

The article needs revisions:

1. Bacterial strains and growth conditions. Did you check real difference between aerobic and semi-aerobic conditions (oxygen content in the medium)?

2. Could you provide any data on strain growth under different conditions? You postulate that the biomass was harvested during mid-log phase. Under different conditions, growth rates, cell densities, growth duration, etc. may differ. Therefore, these data are should be shown.

3. Results and discussion – use MDPI template for Section Titles.

4. Despite the description of the results obtained, discussion of the results obtained is needed.

Briefly, this discussion should include:

- possible biological role of expression differences observed under different conditions;

- comparison of the results obtained with those obtained by other authors.

Author Response

Thank you very much for your thoughtful comments and recommendations. We revised the manuscript based on your comments and recommendations. Specific points are below. Please note that changes are highlighted in red in the revised manuscript.

Reviewer 2

The results of the study may be used for understanding global expression regulation mechanisms in bacteria. The article needs revisions:

Comment 1: Bacterial strains and growth conditions. Did you check real difference between aerobic and semi-aerobic conditions (oxygen content in the medium)?

Response 1: We monitored the growth of the strains by measuring their growth curves under aerobic and semi-aerobic conditions (with and without sulfide) (Pages 86–93). The growth curves have been newly included in the supplemental figure (Supplemental Fig. S1), which shows that the doubling time of the strains under aerobic conditions was approximately 33 minutes. In contrast, the doubling time under semi-aerobic conditions (both with and without sulfide) was approximately 119 minutes. These results indicate that the growth of the strains was slowed by oxygen limitation under semi-aerobic conditions.

Comment 2: Could you provide any data on strain growth under different conditions? You postulate that the biomass was harvested during mid-log phase. Under different conditions, growth rates, cell densities, growth duration, etc. may differ. Therefore, these data are should be shown.

Response 2: The exact growth conditions are carefully described in the Materials and Methods section (Pages 86–93). As mentioned above, we have also newly included growth curves for all strains under all tested growth conditions in the revised manuscript (Supplemental Fig. S1). This additional information should help readers better understand the differences between the growth conditions examined in this study.

Comment 3: Results and discussion – use MDPI template for Section Titles.

Response 3: Thank you pointing out this. We corrected in the revision (Page 114).

Comment 4: Despite the description of the results obtained, discussion of the results obtained is needed. Briefly, this discussion should include:

- possible biological role of expression differences observed under different conditions;

- comparison of the results obtained with those obtained by other authors.

Response 4: We agree with the comment. In the revised manuscript, we have added sentences describing the possible biological roles of the differential expression (Pages 286–296) and sentences comparing our results with those of previous studies (Pages 297–303).

Reviewer 3 Report

Comments and Suggestions for Authors

The authors have described some findings regarding the control of sulfide-responsive transcriptions in Escherichia coli.

General. The authors submitted the manuscript as a brief report, but the submission 13 pages. This is inconsistent. A brief report should be 5 to 6, maximum 7, pages. Hence the authors must significantly decrease the length of the manuscript or else should modify the type of submission.

The authors must make clear the advantages of their approach over similar studies previously published. The literature in general is rich, so the authors should justify their submission.

M & M

Please describe clearly the control strain used in the experimental work.

Presentation of results. The authors are commended for the excellent figures. However, the lack of tables reduces the flow of reading, hence I recommend to reduce the text and increase the tables within the text.

Conclusion. Please note that the concluding section is not in line with the findings. As it is, it reads rather optimistic. Hence, I suggest to tone it down and do not extrapolate.

Overall. The manuscript must be reconsidered after making the above changes.

Author Response

Thank you very much for your thoughtful comments and recommendations. We revised the manuscript based on your comments and recommendations. Specific points are below. Please note that changes are highlighted in red in the revised manuscript.

Reviewer 3

The authors have described some findings regarding the control of sulfide-responsive transcriptions in Escherichia coli.

Comment 1: General. The authors submitted the manuscript as a brief report, but the submission 13 pages. This is inconsistent. A brief report should be 5 to 6, maximum 7, pages. Hence the authors must significantly decrease the length of the manuscript or else should modify the type of submission.

Response 1: We agree for the comment. We here submit the revised manuscript as a regular article.

Comment 2: The authors must make clear the advantages of their approach over similar studies previously published. The literature in general is rich, so the authors should justify their submission.

Response 2: We agree for the comment. In the revised manuscript, we added sentence mentioning the advantages of this study in the Introduction section (Page 68~71).

Comment 3 (M & M):

Please describe clearly the control strain used in the experimental work.

Response 3: In the revised manuscript, we added detailed information of the E. coli strain used in this study (Page 75~77)

Comment 4 (Presentation of results): The authors are commended for the excellent figures. However, the lack of tables reduces the flow of reading, hence I recommend to reduce the text and increase the tables within the text.

Response 4: According to the comment, we tried to reduce text and increase the table as suggested by the reviewer. However, since the figures are schematic diagrams, it was not feasible to present them as tables. We still believe they are best displayed as figures and kindly ask the reviewer to understand our reasoning.

Comment 5 (Conclusion): Please note that the concluding section is not in line with the findings. As it is, it reads rather optimistic. Hence, I suggest to tone it down and do not extrapolate.

Response 5: Thank you for the comment. In the revised manuscript, we have modified the conclusion section in response to the comments from another reviewer (Reviewer 1). Specifically, we added sentences describing the possible biological roles of the differential expression (Pages 286–296) and those comparing our results with previous studies (Pages 297–303). We believe that the discussion section is now described more rationally.

Comment 6: Overall. The manuscript must be reconsidered after making the above changes.

Response 6: We believe that the manuscript is now suitable for publication in this journal.

Round 2

Reviewer 2 Report

Comments and Suggestions for Authors

Authors improved the manuscript. Thus, it may be accepted.

Reviewer 3 Report

Comments and Suggestions for Authors

All issues have been addressed. No further comments.